# Modulation of microRNAs through Lifestyle Changes in Alzheimer’s Disease

**DOI:** 10.3390/nu15173688

**Published:** 2023-08-23

**Authors:** Paola Pinto-Hernandez, Juan Castilla-Silgado, Almudena Coto-Vilcapoma, Manuel Fernández-Sanjurjo, Benjamín Fernández-García, Cristina Tomás-Zapico, Eduardo Iglesias-Gutiérrez

**Affiliations:** 1Department of Functional Biology, Physiology, University of Oviedo, 33006 Asturias, Spain; pintopaola@uniovi.es (P.P.-H.); juan.cas.cas@gmail.com (J.C.-S.); cotoalmudena@uniovi.es (A.C.-V.); manufsanjurjo@gmail.com (M.F.-S.); tomascristina@uniovi.es (C.T.-Z.); 2Health Research Institute of the Principality of Asturias (ISPA), 33011 Asturias, Spain; fernandezbenjamin@uniovi.es; 3Department of Morphology and Cell Biology, Anatomy, University of Oviedo, 33006 Asturias, Spain

**Keywords:** cognitive impairment, epigenetic regulation, lifestyle interventions, physical exercise, diet

## Abstract

Lifestyle factors, including diet and physical activity (PA), are known beneficial strategies to prevent and delay Alzheimer’s disease (AD) development. Recently, microRNAs have emerged as potential biomarkers in multiple diseases, including AD. The aim of this review was to analyze the available information on the modulatory effect of lifestyle on microRNA expression in AD. Few studies have addressed this question, leaving important gaps and limitations: (1) in human studies, only circulating microRNAs were analyzed; (2) in mice studies, microRNA expression was only analyzed in brain tissue; (3) a limited number of microRNAs was analyzed; (4) no human nutritional intervention studies were conducted; and (5) PA interventions in humans and mice were poorly detailed and only included aerobic training. Despite this, some conclusions could be drawn. Circulating levels of let-7g-5p, miR-107, and miR-144-3p were associated with overall diet quality in mild cognitive impairment patients. In silico analysis showed that these microRNAs are implicated in synapse formation, microglia activation, amyloid beta accumulation, and pro-inflammatory pathways, the latter also being targeted by miR-129-5p and miR-192-5p, whose circulating levels are modified by PA in AD patients. PA also modifies miR-132, miR-15b-5p, miR-148b-3p, and miR-130a-5p expression in mice brains, which targets are related to the regulation of neuronal activity, ageing, and pro-inflammatory pathways. This supports the need to further explore lifestyle-related miRNA changes in AD, both as biomarkers and therapeutic targets.

## 1. Introduction

Demographic ageing is one of the main challenges for Western societies nowadays, as it impacts on many levels. According to United Nations, it is predicted that 1.5 billion people will be over 65 years by 2050 [1]. During this period of life, the decrease in the integrity of physiological functions may favor the development of one or more non-communicable pathologies, such as cancer, cardiovascular and metabolic diseases, or dementia. Both age and genetic factors are related to the development of these conditions and are, therefore, considered non-modifiable risk factors. However, the importance of modifiable factors in the development of these conditions, such as diet or physical activity (PA), has become increasingly indisputable. In addition to much scientific evidence supporting lifestyle changes as a prevention and/or treatment for many metabolic and cardiovascular diseases, a great deal of research has also shown a strong link between the development of dementia and lifestyle-related risk factors, like poor dietary habits, physical inactivity, or tobacco and alcohol consumption [2].

### 1.1. Alzheimer’s Disease Pathophysiology

Dementia, a syndrome that affects the brain in a primary or secondary way, results in a progressive and chronic loss of cognitive function, more abruptly than what occurs during physiological ageing, and is the leading cause of disability in the elderly population worldwide [3]. Alzheimer’s disease (AD) is the most common disease within dementia, comprising 60–70% of cases [2]. In more than 98% of the cases, AD has a sporadic and late onset development (LOAD), while only 1–2% rely on a genetic origin. It is considered that older age, the *APOE4* allele, and a family history of AD are the greatest risk factors for this condition [2]. 

The onset of the first alterations underlying this disease appear in a long pre-symptomatic phase, up to 10 years before the onset of the initial symptoms. This slow and silent progression to a stage where the initial symptoms begin to emerge is often referred to as the AD continuum [2], summarized in Figure 1. Once the damage produced by this asymptomatic phase has already had a functional effect, the progressive development of symptoms begins. Subsequently, the degeneration spreads to different brain structures, leading to the progressive development of symptoms, from mild cognitive impairment (MCI) due to AD to a severe state of dementia with very pronounced symptomatology [4].

The neuropathology of AD is first characterized by the extracellular accumulation of amyloid plaques consisting of Aβ_1–40_ and Aβ_1–42_ peptides generated by amyloid precursor protein (APP) cleavage by β-site APP-cleaving enzyme 1 (BACE1) and the γ-secretase complex. This is then followed by intracellular aggregation of the hyperphosphorylated tau protein, developing neurofibrillary tangles [5].

Most research to delay AD focuses on Aβ clearance, because it is widely regarded as a pathological trigger [5,6]. This clearance can occur via three different pathways involving different compartments, including intracellular and extracellular clearance and outflow from the central nervous system (CNS) into the systemic pathway [7,8,9,10]. Intracellularly, Aβ and APP structures are encapsulated into autophagic vesicles (autophagosomes) in the neuronal axon and then, by retrograde transport to the neuronal body, are degraded in lysosomes. It seems that retrograde transport can be disrupted in AD by dynein–dynactin complex dysfunction because of hyperphosphorylated tau. Thus, this produces a pull of immature autophagosomes loaded with Aβ on the growth cone, leading to neuronal function loss and neurodegeneration [11,12]. Extracellularly, microglia collect and clean up soluble Aβ species, phagocyte insoluble fibrillar Aβ deposits, and induce both the activated microglia state and the trapping of amyloid plaques [8]. However, microglia appear to play a dual role in the clearance of these plaques. At the early stages, when the plaques are small, anti-inflammatory phenotype M2 microglia form a protective barrier around fibers, compacting them and preventing the binding of more protein, in order to control the neuroinflammation. However, Aβ oligomers promote the pro-inflammatory microglia phenotype M1. This leads to its chronic activation, contributing to neurotoxicity and synaptic loss [13]. In addition, soluble Aβ can move through the blood–brain barrier (BBB) both into and out of the CNS from the systemic circulation. This requires specific transporters, such as the receptor for advanced glycation end products (RAGE) and the lipoprotein receptor-related protein 1 (LRP1), respectively [14]. Thus, a change in the balance of these two pathways, along with the loss of BBB integrity, due to the pathophysiology of AD, may contribute significantly to the accumulation of Aβ within the CNS [15,16,17].

This chain of events taking place at brain level can be prevented or delayed by acting on modifiable risk factors. In fact, adherence to a healthy lifestyle is associated with a 60% lower risk of LOAD [18], with the top risk factors being obesity in middle age and a sedentary lifestyle [19]. Thus, it would be necessary to understand how diet and PA can help to enhance those pathways that favor Aβ clearance and decrease the risk of neuroinflammation in the CNS. Therefore, a healthy lifestyle could help in the prevention of AD, delaying its onset at the preclinical phase, and slowing down disease progression during MCI, its prodromal stage.

### 1.2. Food Choices and Eating Habits Impact AD Pathophysiology

Growing evidence supports the protective role of dietary factors against cognitive impairment and healthy ageing [20]. In particular, the Mediterranean diet (MedDiet) was among the first to be associated with benefits to cognitive function, later joined by heart-healthy diets, such as DASH (Dietary Approaches to Stop Hypertension), which finally merged into the so-called MIND diet (Mediterranean-DASH Intervention for Neurodegenerative Delay) [21,22]. 

Likewise, several studies have reported that an unhealthy diet and the consumption of certain foods may trigger the development and progression of AD [23,24]. Similarly, some dietary components have been described that may help to reduce the accumulation or favor the elimination of Aβ, as well as processes that support the maintenance of brain health (Table 1). More specifically, polyphenolic compounds have been associated with anti-amyloidogenic and anti-aggregation functions, reducing the Aβ aggregation kinetics involved in the pathophysiology of AD [20,25]. In addition, several studies have shown that omega-3 fatty acid intake affects Aβ clearance, through increasing LRP1 expression [26] and autophagy [27], and reducing neuroinflammation and neuronal loss [28], which could attenuate AD pathology [29]. In contrast, cholesterol-rich diets and red and processed meat may contribute to the accumulation of Aβ, in the latter case due to its high content of trivalent iron, which induces the formation of fibrin fibers that interact with Aβ [29]. Concerning the neuroinflammation process during the disease, phenolic acids have neuroprotective effects by acting on neuroinflammatory pathways, decreasing the level of pro-inflammatory mediators, and increasing the production of neurotrophic factors, such as brain-derived neurotrophic factor (BDNF) [20]. 

To sum up, diets rich in vegetables, fruits, legumes, nuts and seeds, moderate in fish consumption, and low in processed cereals, red meat, and saturated fats, in line with the MedDiet model, which also include the use of olive oil as the main culinary fat, are appropriate strategies that can help to reduce the risk of developing AD, more effectively as a preventive strategy than when the disease has already started [30,31,32,33].

### 1.3. Physical Activity as a Protective Factor in AD Pathophysiology

Numerous reports published by the WHO, Alzheimer’s Disease International, and other health organizations have proposed lifestyle changes, mainly PA, to prevent and/or delay the onset of dementia in the older population [2,34,35,36,37]. PA has been related with the maintenance of cognitive function, even in MCI patients [2,38], and with a better subjective quality of life [39].

A recent review has shown great heterogeneity in the literature regarding volume, frequency, intensity, and duration of exercise interventions in AD patients [40]. Furthermore, most protocols include endurance training and/or sessions, while the study of resistance exercise and concurrent training interventions in these patients have received less attention [41,42]. However, PA interventions (two to three 30–60-min sessions per week, maintained for two months), combining endurance and muscle strengthening exercises, may improve some markers of cognitive function, neuropsychiatric symptoms, functional independence, and physical function in patients with AD [40]. In terms of the benefits at the brain level, exercise promotes the efficient utilization of neural networks, an increased recruitment of connections, and an improved cognitive reserve, both in the pre-symptomatic and the pre-dementia phases [43,44,45,46]. At the molecular level, studies using AD murine models indicate that exercise ameliorates the pathology, acting on many of the pathways dysregulated on AD. Thus, at the intracellular level, exercise can both reduce the production of Aβ from its precursor and increase its elimination through autophagy [47,48], reducing the potential accumulation of Aβ deposition [7,49]. At the extracellular level, a decrease in the activation of microglia and astrocytes, thus improving neuroinflammatory processes, has been observed [50,51,52]. Moreover, exercise modifies the flux of soluble Aβ through the BBB, decreasing its entry from the systemic circulation, by reducing the levels of RAGE [50], and increasing the levels of LRP1, favoring Aβ exit from the CNS [52]. Indeed, it is noteworthy that the effect of exercise on LRP1 is dose-dependent [53]. In addition to improving Aβ clearance, other benefits of exercise on the pathophysiology of the disease have been described. These include the increase in some neurotrophins, such as BDNF, which may affect brain plasticity in the young and adults during physiological ageing [54], as well as the enhancement of neurogenesis in the adult hippocampus [55,56]. Interestingly, in the case of MCI patients, one of the most striking effects of PA on the brain is the prevention of hippocampal volume loss [38].

### 1.4. Molecular Mechanisms Underlying the Protective Effect of Lifestyle on AD: The Role of microRNAs

Although some of the benefits of both diet and PA on AD have been demonstrated at the molecular level, much remains to be known about the underlying molecular mechanisms by which this protective effect is exerted. Among them, the epigenetic regulation mediated by non-coding RNAs, particularly microRNAs (miRNAs), has received much attention in recent years in the etiology and pathogenesis of AD [57,58,59].

MiRNAs are short non-coding RNA molecules (20–22 nucleotides) that post-transcriptionally regulate gene expression by promoting mRNA degradation or inhibiting its expression [60]. MiRNAs play a key role in a multitude of biological processes, as a miRNA can have a multitude of mRNA targets and the same mRNA can be targeted by several miRNAs. They are usually located intracellularly, and tissue-specific profiles have been described [61]. However, they can also be secreted to different biological fluids, constituting the so-called circulating miRNAs (c-miRNAs).

The study of the role of miRNAs as modulators of the pathophysiological processes that define AD [62], as well as their role as biomarkers of disease onset and progression has strongly emerged recently [63,64]. In fact, numerous miRNA gene targets have been described and validated in AD-associated pathways, such as the APP processing pathway [65,66], autophagy [67], synapsis [68,69], neuroinflammation [70,71], and cognitive function [72,73]. Even more, several articles have described the dysregulation of some c-miRNAs in the serum/plasma and the cerebrospinal fluid (CSF) of AD patients [74,75]. The miRNA expression in humans has also been carried out from postmortem brain tissue samples [76,77]. Although the results between studies are heterogeneous, Improta-Caria et al. [78] showed that several miRNAs exhibited significant dysregulated expression in both brain and blood in AD subjects. For instance, miR-181c-5p and miR-29c-3p, downregulated in the brain and upregulated in blood, and miR-125b-5p, miR-146a-5p, and miR-223-3p, upregulated in the brain and downregulated in blood. Some of these miRNAs are also involved in several dysregulated pathways during the disease in patients. For instance, decreased levels of miR-146a in brain are related to neuroinflammatory processes [78], while decreased levels of the miR-29 family (including miR-29a, miR-29b-1, and miR-29c), which regulate BACE1, favor Aβ production and accumulation, as observed in AD [78]. Additionally, it was also described that Tau hyperphosphorylation is related to the overexpression of miR-125b in AD brain samples [78]. 

Interestingly, both diet and PA can modulate miRNA expression and it is suggested that they can provide their benefits in many human pathologies through the regulation of miRNAs [79,80,81,82,83,84]. Some evidence supports that a wide range of nutrients and bioactive compounds can alter the expression of specific miRNAs [85,86,87,88,89]. Moreover, acute exercise and training elicit changes in the expression, type, and circulation appearance–clearance kinetics of certain c-miRNAs in both healthy and diseased individuals [90] and are able to alter brain miRNA abundance in animal models [79,80,81,82,83,84]. However, there is still little information on the impact of diet and PA on the regulation of miRNAs in AD. Furthermore, the study of miRNAs in AD patients has mainly focused on plasma c-miRNAs, which are easily accessible in vivo, while little information is available about what happens in tissues, particularly the brain. This limitation can be overcome by using AD mouse models (Table 2), which allow the study of miRNA expression in brain tissues, although their direct translation to patients is limited. 

However, based on the knowledge that can be gained from studies in AD patients and mouse models, it is hoped that it will be possible to establish whether changes in the expression levels of key miRNAs induced by diet and/or PA, can modify both disease onset and/or progression (Figure 2).

Therefore, the aim of this review has been to compile and to critically analyze the available information on the modulatory effect of diet and PA on the expression of miRNAs described as dysregulated in the context of AD. In this way, we will be able to establish critical points for further progress in the development of strategies to: (i) explore the value of miRNAs as prognostic biomarkers in MCI and AD, taking into account the modulatory effect of lifestyle; (ii) implement non-pharmacological, cost-effective, targeted and personalized interventions, whose effect on patients’ risk trajectories can be monitored; and (iii) identify new therapeutic targets that allow the development of new treatments for this devastating disease.

## 2. Regulation of Altered miRNAs in AD Pathophysiology by Diet

There are many studies suggesting the modulation of altered miRNA expression in different non-communicable diseases through diet or dietary factors [91,92]. However, when trying to identify studies that analyze the impact of dietary patterns, nutrient intake, or food bioactive compounds on miRNA expression regulation in MCI or AD patients or even in AD murine models, we have found that these studies are really scarce. Table 3 summarizes the studies that have analyzed the relationship between nutrition and miRNA expression, at both plasma and brain levels, in humans and mouse models, respectively.

### 2.1. Human Studies

Regarding studies in patients, Zhang et al. conducted an ambitious observational study in a cohort of 75 MCI patients in the pre-dementia stage of Alzheimer’s disease (AD), and 52 healthy controls of both sexes [93]. These participants were obtained from the Effects and Mechanism Investigation of Cholesterol and Oxysterol on Alzheimer´s disease (EMCOA) study [95]. In this case, their aim was to determine the interactive role between diet quality, serum miRNA levels, and gut microbiota to discriminate MCI subjects from healthy subjects. Global cognitive function was assessed by two standard cognitive screening tests, the Mini-Mental State Examination (MMSE) and the Montreal Cognitive Assessment (MoCA). Those participants suspicious of having MCI were further examined by neurologists to confirm the final diagnosis. Diet quality was comprehensively recorded by the Chinese Dietary Guidelines Index 2018 (CDGI-2018), by the Energy-Adjusted Dietary Inflammatory Index (E-DII), and by the healthy lifestyle score (HLS). The CDGI-2018 consists of 13 components, including the percentage of energy intake from carbohydrates, whole grains and mixed beans, fruit, total vegetables, the proportion of dark vegetables in total vegetables, soybean and nuts, dairy products, meat and poultry, eggs, aquatic products, oil, salt, and alcohol. The E-DII was used to assess the inflammatory potential of diet. Therefore, eight pro-inflammatory parameters (total energy, carbohydrate, protein, total fat, saturated fatty acids, cholesterol, vitamin B_12_, and iron) and sixteen anti-inflammatory parameters (alcohol, MUFAs, PUFAs, fiber, thiamine, riboflavin, niacin, folic acid, vitamin B_6_, vitamin A, β-carotene, vitamin C, vitamin E, zinc, magnesium, and selenium) were included in this index. The HLS includes scores from five lifestyle factors coded as healthy (+1) or unhealthy (0): diet, smoking, alcohol consumption, physical activity, and BMI. Their results showed lower scores on the CDGI-2018 and HLS in MCI participants, with no differences in E-DII [93].

For the study of serum c-miRNAs, they performed a qPCR analysis of a small panel of four miRNAs: hsa-let-7g-5p, hsa-miR-107, hsa-miR-144-3p, and hsa-miR-186-3p. This selection was based on their previous results, obtained in a similar MCI cohort [96]. In this previous study, they used an array containing 2578 mature miRNAs to analyze specific miRNA profiles in five MCI patients and five healthy matched individuals per group, finding twenty deregulated miRNAs in MCI patients. These miRNAs were selected based on the following criteria: >1.2-fold change and adjusted *p*-value < 0.05 [96]. From these results, the authors carried out an ulterior validation using qPCR in 50 MCI patients and 50 healthy controls. Surprisingly, the validation was carried out only in four out of twenty deregulated miRNAs, two upregulated (hsa-miR-144-3p and hsa-miR-186-3p), and two downregulated (hsa-miR-107 and hsa-let-7g-5p) miRNAs [96]. Unfortunately, the authors do not justify why they specifically selected these four miRNAs for validation, and neither is it evident in view of their fold-change nor their *p*-value. When these c-miRNAs were analyzed in the MCI cohort described in Table 3 [93], hsa-miR-186-3p, hsa-miR-107, and hsa-let-7g-5p were significantly repressed in the MCI group compared to their aged-matched healthy controls, while hsa-miR-144-3p levels were higher. Strikingly, the repression of miR-186-3p in MCI was, again, inconsistent with the overexpression or the absence of differences previously reported by the same authors [96].

With the information collected on diet, c-miRNA profile, and gut microbiota, the authors performed a multivariate logistic analysis [93]. Interestingly, the multivariate regression models used showed that diet quality scores alone were not able to distinguish between MCI and control participants, while serum miRNAs alone showed a low potential for this discrimination. A better predictive value was observed by a model which included all the diet quality scores, the four miRNAs analyzed, and several bacterial taxa found in fecal samples [93]. Surprisingly, the predictive value of diet quality indexes and c-miRNA levels together were not shown. Therefore, the relevance of diet and its relationship with c-miRNA levels in MCI was not evident considering these results [93]. However, certain components of the diet have been associated with better cognitive function and reserve in MCI and AD patients, particularly folates and vitamin B_12_ [97], although results show high heterogeneity, and the underlying mechanisms are not clear.

### 2.2. Mouse Studies

In order to further investigate whether certain dietary components could have a beneficial effect during pathology at the brain level, Liu et al. [94] examined the effect of a folic-acid-deficient diet in the APP/PS1 AD mouse model (Table 3). These mice express human transgenes for both *APP* and *PSEN1* with known familial AD mutations and develop only Aβ plaques in the brain [98]. Thus, these authors performed a 60-day intervention where 7-month-old APP/PS1 mice were assigned to two different groups: folic-acid-deficient diet (0.2 mg folic acid/kg diet) and control diet with normal folic acid content (2.1 mg folic acid/kg diet). They found that following a folic-acid-deficient diet exacerbated spatial memory impairment, although not affecting learning performance, as evaluated by the Morris water maze test. In addition, soluble levels of Aβ_1–42_ were increased in folic-acid-deficient mice, regarding APP/PS1 under control diet, as well as the mice showing increased levels of Aβ_1–42_ deposits in the cerebral cortex and hippocampus. Furthermore, this dietary intervention significantly decreased the expression of three miRNAs in brain: miR-106a-5p, miR-200b-3p, and miR-339-5p. Bioinformatic analysis showed that a potential target of miR-106a-5p and miR-200b-3p is *APP*, while *BACE1* is a potential target of miR-339-5p. In fact, APP and BACE1 protein levels were increased in APP/PS1 mice deficient in folic acid [94]. However, this study did not determine whether these miRNAs were also altered in the plasma of these mice, missing the opportunity to determine whether the dysregulation observed in the brain could be detected as plasma c-miRNAs.

Therefore, further studies are needed, overcoming the limitations highlighted above, to better understand how the diet, or certain nutrients, may regulate the expression of specific miRNAs in the context of neurodegenerative diseases.

## 3. Exercise Modulates the Expression of miRNAs Altered in AD

Despite the known positive effect of exercise on AD prevention and progression, there is still little information on the underlying molecular mechanisms. It is especially limited in terms of the modulatory role of exercise on miRNA expression, both in AD patients and in murine models of the disease. Table 4 shows the most relevant information about the studies that have addressed this question and the main results obtained.

### 3.1. Human Studies

Two studies have analyzed the effect of endurance exercise on serum c-miRNA levels in AD patients. In both cases, the training protocol was based on three months of cycling at a maximum heart rate of 70% [99,100], with no further data on how this protocol was implemented. Thus, it would have been desirable to have more information about the days per week, time and intensity per session, adherence to the intervention, etc. Since these variables have been described as highly influencing c-miRNA profiles in young males [90], it cannot be ruled out that they also influenced this study.

Additional weaknesses in the experimental design were found. First, the lack of a matched control group of healthy subjects. Both studies included only AD patients who were divided into two groups: exercise and non-exercise (Table 4). This makes it difficult to know whether the changes observed in the response of c-miRNAs to the training protocol is similar to what might be observed in exercising healthy individuals or, even, whether exercise in patients induces a profile more similar to that of healthy individuals. In addition, both men and women were recruited but considered together, so the eventual gender-specific effect of the exercise intervention is not known. Furthermore, in each study, the authors measured only one specific c-miRNA in serum [99,100], before and after the intervention period. In the case of Qin et al. [99], they determined the levels of miR-192-5p, which has been previously reported to be abnormally expressed in aged people and to be a key regulatory molecule associated with AD [106,107]. Although no difference for serum miR-192-5p expression was found between both groups at the beginning of the intervention, after 3 months of training the exercised AD patients showed a significant downregulation of this miRNA, which was not observed in the non-exercised AD patients [99]. More importantly, the exercise intervention ameliorated cognitive impairment in these patients. The study of Li et al. was focused on miR-129-5p, which has been reported to inhibit neuroinflammation in ischemia-reperfusion injury in the spinal cord [100]. After 3 months of endurance training, serum miR-129-5p expression was significantly upregulated in the exercise group compared to the control group [100]. Therefore, in both studies, the authors found that the different response in the expression of these c-miRNAs was associated with a decrease in the levels of the pro-inflammatory cytokines TNF-α, IL-6, and IL-1β [99,100].

In conclusion, both studies showed that exercise can modulate different miRNAs in AD patients, even in opposite directions, but affecting the same downstream routes, in this case, the pro-inflammatory response associated with AD. This is an interesting finding, as it implies that a set of miRNAs could be dysregulated, in opposite directions, but affecting the same physiological pathway.

### 3.2. Mouse Models

Although the studies discussed above showed how PA can modulate the expression of c-miRNAs in AD patients, it is not yet possible to study in vivo whether PA is able to change the miRNA expression in different brain regions involved in the pathology in humans and to generate hypotheses about which integrated pathway clusters may be involved in the disease at this level. Again, mouse AD models are useful to address these questions. In fact, there are several studies that have approached this aspect, but huge differences were observed among them in the training protocol used, the sex and age of the mice, the brain area analyzed, and the AD mouse model used, as shown in Table 4.

Among the seven studies that examined the effect of exercise on the miRNA profile in the brain by means of murine models (summarized in Table 2), those based on the accumulation of Aβ are more common, such as the APP/PS1 mouse model [99,100,101] and the 5xFAD model, the latter characterized by a faster progression. In addition, other murine models of the disease were used regarding both plaque and tangle pathology such as 3xTg-AD [103]. Apart from these genetic models, two studies [104,105] used a mouse model of accelerated spontaneous senescence (SAMP8), which spontaneously developed an older age phenotype and a shorter life expectancy, as a disease model.

Although they reflect many of the hallmarks of the human pathology, all these AD murine models show differences at the phenotypic level. Thus, in order to group the studies on the basis of a common factor and considering that the aim of this review is to study how lifestyle can modify dysregulated miRNAs during disease, we chose the training protocol as a criterion. Most authors have investigated the effects of endurance exercise, by means of a voluntary wheel running (VE), although with different training periods (Table 3). The most common duration of endurance exercise interventions ranged from 4 weeks [99,104] to 8 weeks [101,102,105]. Another thing in common in these studies is that almost all of them focus on the study of a single miRNA in the hippocampus, one of the brain regions most affected by AD.

Four of these studies used murine AD models that accumulate and form Aβ plaques [99,100,101,102]. However, one of them failed to observe changes in the selected miRNA, miR-128, in the hippocampus of the 5xFAD model after 8 weeks of VE [102]. Two of the studies were executed in parallel with the corresponding analysis of a single c-miRNA in AD patients who also underwent an endurance exercise intervention, miR-192-5p and miR-129-5p [99,100]. Interestingly, in both, the 4-week VE intervention in the mouse model APP/PS1 modified the expression of the selected miRNA in the hippocampus. The observed changes were in the same direction detected in AD patients, i.e., a decrease in miR-192-5p levels [99], and an increase in miR-129-5p levels [100]. Moreover, both miRNAs were shown to be implicated in the regulation of inflammatory cytokines, such as TNF-α, IL-6, and IL-1β in the hippocampus. These results were further supported by a subsequent study where another anti-inflammatory miRNA, miR-130a-3p, was found to increase its expression in the hippocampus of APP/PS1 after 8 weeks of VE [101]. Thus, exercise modulation of miR-192-5p, miR-129-5p, and miR-130a-3p reduces the expression of pro-inflammatory cytokines, decreasing neuroinflammation. It was also noteworthy that miR-129-5p knockdown reduces the protective effects of PA against cognitive dysfunction in AD mice [99,100], which reinforces the specificity of this intervention at this level.

Another interesting study was conducted by Dungan et al. [103], as it presents many differences in design from the previous ones. They used the 3xTg-AD mouse model, using a 20-week intervention, where the mice in the exercise group performed a combined endurance and resistance protocol, called progressive weighted wheel running (PoWer). Moreover, they evaluated an miRNA expression panel, selecting those miRNAs that have *Bace1* as the predicted or validated target. In this way, they observed an increase in miR-29a, miR-29b, miR-29c, miR-107, miR-328, miR-129, and miR-140 in the hippocampal region of the PoWeR group. It is remarkable that, with this strategy, miR-129 was also detected, in the same way that it was found in the APP/PS1 study [100]. Therefore, only a few studies have been completed so far in murine models of AD and, regardless of the differences between training protocols, AD model, age, and sex used, all of them show the existence of exercise-modulated miRNAs associated with AD molecular pathways.

Since the main risk for LOAD development is age, we have also considered two studies evaluating VE in a senescence-accelerated murine model without genetic modifications, SAMP8. This model presents deficits in learning and memory where the hippocampus plays an important role. Thus, one of the studies analyzed hippocampal samples from 8-week VE-trained SAMP8 female mice, by means of a commercial panel with 84 miRNAs associated with neurodevelopment and disease [105]. Their results highlight the increase in miR-28-5p, miR-98-5p, miR-148b-3p, miR-7a-5p, and miR-15b-5p, and the decrease in miR-105 and miR-133b-3p. In the case of the second study, a 4-week VE intervention was performed in male SAMP8 [104]. In contrast, they only analyzed one miRNA, miR-132, involved in the regulation of neuronal activity, albeit in several brain regions. Interestingly, their results showed that miR-132 was only increased after VE intervention in the hippocampus [104]. Moreover, they observed that VE also decreases levels of APP, the Aβ precursor.

Hence, all the studies considered, whether in AD or in ageing mouse models, confirm that the hippocampus is a brain region that shows a response to PA through the modulation of miRNAs whose dysregulation affects molecular pathways associated with AD. However, once again, the analysis of these dysregulated miRNAs at plasma or serum level has not been considered, missing important information that could help to establish new biomarkers of the pathology using relatively accessible markers.

## 4. Pathway Enrichment Analysis of miRNAs Modulated by Lifestyle Interventions during AD in Humans

Unlike animal studies, it is not possible to analyze how lifestyle may modify miRNA expression at the brain level in AD or MCI patients, except in samples collected postmortem. Therefore, the studies available and included in this review merely describe the differential profile of c-miRNAs in relation to diet and exercise. Unfortunately, a functional analysis in vitro or in silico of the c-miRNAs described has not been addressed in any of them. This makes it difficult to understand in which integrated metabolic pathways these miRNAs might be involved and to deepen in their regulatory role. This information is relevant to assess the molecular effect of lifestyle interventions in humans and the potential regulatory role of these miRNAs in the context of AD or MCI.

Therefore, we have performed an in silico analysis of those miRNAs described as deregulated in the human studies included in Table 2 and Table 3. On the one hand, we jointly analyzed let-7g-5p, miR-107, and miR-144-3p, the three miRNAs consistently repressed in MCI patients with a low adherence to dietary recommendations, as described by Zhang et al. [93,96]. Then, we performed a similar analysis for those miRNAs that were modified after an exercise intervention in AD patients: miR-129-5p [99] and miR-192-5p [98].

To describe the metabolic pathways in which let-7g-5p, miR-107 and miR-144-3p are implicated, we performed an exploratory in silico analysis using DIANA-miRPath v.4.0 integrated with TargetScan v.8.0 and KEGG pathways. Figure 3 shows this analysis, where it can be seen a strong association of these three miRNAs with the following processes (excluding processes related to cancer and infectious diseases): axon guidance, signaling pathways regulating pluripotency of stem cells, MAPK signaling pathway, Wnt signaling pathways, Hippo signaling pathways, PI3K-Akt signaling pathways, and TFG-beta signaling pathway, among others.

Some of the pathways identified have been previously studied in the context of AD. For example, Axon-guidance molecules, which include netrins, ephrins, Slits, and semaphorins, play an important role in guiding growth cones to form synapses and are involved in the pathogenesis of AD by regulating the levels of Aβ and the hyperphosphorylation of tau through various signaling pathways [108]. In fact, netrin-1 has been shown to be influenced by diet, since high levels of methionine and low levels of folates and vitamins B_6_ and B_12_ can affect the methylation levels of the *Netrin-1* gene promoter, downregulating its expression and favoring memory loss (reviewed in [108]). The Hippo signaling pathway has been shown to affect the production of pro-inflammatory cytokines associated with microglial activation during AD pathogenesis (reviewed in [109]). The PI3K-Akt signaling pathway is one of the most studied in AD, as GSK-3β, one of its downstream effectors, shows increased activity that is directly associated with the enhanced production and deposition of Aβ and the hyperphosphorylation of tau that occurs during the disease [110]. Lastly, regarding other pathways such as the TGF-beta pathway, the deficiency in this growth factor has been shown to increase both Aβ accumulation and Aβ-induced neurodegeneration in AD models. The loss of function in the TGF-beta pathway seems also to contribute to tau pathology and neurofibrillary tangle formation [107]. Therefore, considering the dysregulation of several of these pathways during AD pathogenesis, miRNAs could act as modulators of several components, potentially revealing their role as therapeutic targets.

In addition to playing an important role in the regulation of these pathways, dysregulation of these three miRNAs has been reported during the progression of pathology. In the case of let-7g-5p, its upregulation in the serum of AD patients makes it a promising biomarker for diagnosis [111,112]. As for miR-107, it has been widely described as playing a role in AD pathophysiology, since it is downregulated in the temporal cortex at Braak stages III/IV of MCI subjects, upregulating BACE1 expression, and increasing the accumulation of Aβ. Similarly, it is downregulated in the temporal cortical grey matter during the early stages of AD [113,114]. Regarding miR-144-3p, the upregulation in serum of MCI patients with a lower adherence to dietary recommendations, described by Zhang et al. [93,96], is supported by its overexpression in AD patients’ postmortem brain samples. The authors describe this miRNA as a negative regulator of the metalloprotease 10 (ADAM10) in the pathogenesis of the disease [115]. This protein is the main α-secretase in the non-amyloidogenic cleavage of amyloidogenic precursor protein (APP), so its activation would impede the amyloidogenic pathway by preventing the production of amyloid-β peptide (Aβ) [116]. To the contrary, Zhou et al. [117] have showed a reduction in this miRNA, miR-144-3p, in AD group [117]. In their article, the authors reported that one of the possible targets of this miRNA is located in the 3′UTR of the *APP* gene. At the same time, they also found that genetic variations in the -534G/A and -118C/A sites in the *APP* 3′UTR gene region affected the binding of miRNAs and, therefore, the regulation of APP expression by miRNAs, highlighting the importance of dysregulation of these genes and the miRNA machinery during this pathology [117].

Similar in silico analysis was performed for miR-129-5p and miR-192-5p, which were modified by exercise interventions in AD patients (Figure 4) [99,100]. This analysis revealed that only miR-129-5p appears to play a role in several metabolic pathways, such as the calcium signaling pathway, Axon guidance, signaling pathways regulating the pluripotency of stem cells, TGF-beta signaling pathway, etc. In contrast, the validated or predicted gene targets of miR-192-5p do not allow the identification of any specific pathway in which it might be involved.

Some of the pathways in which miR-129-5p participates are consistent with the previous in silico analysis (Figure 3), such as the signaling pathways regulating the pluripotency of stem cells, Axon guidance, Hippo signaling pathways and TGF-beta signaling pathway, revealing common pathways related to neurodegenerative pathology, involving different miRNAs that respond to lifestyle-related factors (diet, exercise). However, this miRNA also appears to play a role in a very relevant pathway in the context of AD pathology, as it is the calcium signaling pathway [118].

Intracellular calcium acts as a second messenger and plays a key function in the regulation of neuronal functions, such as neural growth and differentiation, action potential, and synaptic plasticity. The calcium hypothesis of AD proposes that the activation of the amyloidogenic pathway affects neuronal Ca^2+^ homeostasis and the mechanisms responsible for learning and memory [118].

In relation to the role of these miRNAs in the pathology of AD, miR-129-5p has been reported to be reduced in AD patients in both the superior temporal gyrus and entorhinal cortex [107], while for miR-192-5p there is not much information on their expression in AD. Only Rahman et al. [107,119] described miR-192-5p as a key signaling and regulatory molecule associated with transcriptional changes and deregulated in some ageing-related disease [107]. This lack of information is consistent with the aforementioned lack of metabolic context for this miRNA observed in the in silico analysis.

Taking the results of the in silico analysis together, the current studies in this review are insufficient to reach a firm conclusion. However, it appears that the miRNAs described are involved in several metabolic pathways that play an important role during disease progression. Nevertheless, considering that the diet studies described here, relating to MCI due to AD patients, do not perform any intervention, it would be interesting to assess whether nutritional interventions, described in other studies in other contexts, modified the expression of some of these miRNAs. miR-107 is one of the miRNAs that has been described to be modified by dietary changes in both humans and mice. For example, Khorraminezhad et al. observed that a high dairy diet (≥4 servings/day according to the 2007 Canadian dietary guidelines) decreased miR-107 expression at the circulating level compared to an adequate dairy diet (≤2 servings/day) for 6 weeks [120]. In mice, the downregulation of this miRNA was also observed after polyphenol administration in the liver of mice fed a high-fat diet compared to mice fed a dairy diet [121].

This would allow the determination of whether a dietary intervention may have an effect on the physiological pathways involved in the disease, as was seen in the AD murine studies [94]. However, due to the complexity of dietary effects and the fact that miRNAs can be regulated by multiple factors, it is difficult to isolate the interaction between them. For this reason, it is necessary to approach the understanding of the effect of diet on miRNA expression from different perspectives, including not only studies in humans or murine models, but also cellular models.

## 5. Discussion

Despite the large number of studies that corroborate that alternative strategies based on multimodal approaches (diet, exercise, and cognitive training) seem to be more promising on AD prevention [45,122,123] and that miRNAs are one of the epigenetic regulators that have been described as dysregulated during the disease [124,125], the information collected in this review is very limited. However, several articles have proposed lifestyle changes, mainly diet and PA, as a preventive strategy to modulate the expression profile of miRNAs due to their essential role as mediators of several processes such as angiogenesis, metabolism, cell proliferation, tumor growth metastasis and invasion, or neuronal regeneration [126,127,128]. Thus, there is still a long way to go before determining whether the regulation of miRNAs, via these modifiable lifestyle factors, could prevent the development of MCI due to AD and, even, whether, once it appears, they can prevent progression to the clinical stage of AD.

We are not aware of any dietary intervention studies with MCI or AD patients in which miRNAs were measured, nor of whole diet modification, nor of administration of any particular nutrient or bioactive compound. Only the study of Zhang et al. [93] evaluated the quality of diet and a restricted c-miRNA panel of MCI due to AD patients. However, although the authors reported a group of miRNAs dysregulated in these patients, which functional analysis revealed a close relationship with pathways dysregulated in AD, they did not establish a relationship with diet quality or their joint predictive value of MCI risk, although a lower diet quality was reported for MCI patients. On the other side, and albeit studies are still scarce, there are two studies where a PA intervention in AD patients was carried out and miRNAs were analyzed. Both considered cycling as an endurance exercise intervention and, surprisingly, none of them included a group of healthy subjects [99,100]. Therefore, it cannot be known whether the response to exercise in the modulation of miRNAs brings AD patients closer to healthy subjects, in terms of miRNA profile. In addition, caution should be also taken regarding the exercise model used for intervention studies. Although endurance exercise has been prescribed more frequently than resistance exercise to improve cognitive health, the latter has also shown many benefits in these patients [129]. In fact, there is a strong rationale that carefully programmed resistance exercise elicits positive neuroplastic adaptations and offers different cognitive benefits than endurance exercise [130]. Furthermore, in terms of miRNA expression, it has been widely described that endurance and resistance exercises induce different c-miRNA profiles and expression levels [131]. Hence, it is expected that the modulatory role of miRNAs in improving cognitive health would be different for endurance and resistance exercise. Therefore, it seems necessary to explore the modulatory effect of resistance exercise on the expression of miRNAs in the context of neurodegenerative pathology.

Moreover, none of the studies with MCI or AD patients, either in relation to diet or exercise, have performed a massive sequencing technique for miRNA analysis, which would expand the number of miRNAs found to be modified by these two factors. Instead, the authors analyze a small panel of c-miRNAs previously described as related to AD pathology, with hardly any coincidences between the different studies in the specific c-miRNAs considered. The use of such restrictive approaches provides an incomplete understanding of both the effect of lifestyle on miRNA expression and on the amplitude of their regulatory role, limiting the possibility of identifying which physiological pathways are regulated by these miRNAs, beyond those defined by disease hallmarks, and closing the door to the discovery of new therapeutic targets. In addition, another common limitation is that, in human studies, the in vivo analysis of miRNAs has been carried out in serum/plasma samples. Obviously, it is not possible to analyze in vivo miRNA expression in human brain tissue. To overcome this, animal models are a good alternative for a translational perspective. In fact, there are several well-characterized murine AD models that express mutated forms of one or more of the genes involved in the pathology. These models make it possible to obtain samples that are otherwise difficult to collect in human studies.

Unfortunately, we have also found some limitations in murine AD model studies that we have observed in those conducted in patients. For instance, applying a restrictive approach, analyzing one or a predefined set of microRNAs in a commercial array, and, most surprisingly, only analyzing miRNAs at the brain level. Thus, the opportunity to explore whether these changes were also detected at the circulating level, either in plasma or in CSF, is lost. This would have opened the door to correlating circulating levels with the changes observed in the tissue. Although we have been able to find studies in murine models of AD in which a diet or PA intervention was performed, followed by miRNA analysis, they are still very scarce. In fact, we have only found one diet intervention study in this regard [94]. As for PA interventions, in most studies mice performed voluntary exercise on a running wheel inside the cage, but their activity was not recorded [99,100,101,102,104,105]. Thus, this is another major limitation, since the training protocol did not quantify the amount of endurance the mice performed. On the other hand, it should also be considered that it would be necessary to evaluate the impact of other exercise models, such as resistance exercise. In addition to all this, there are still differences in terms of the AD model used, age, and sex of the mice used in these PA intervention studies, which may introduce even more factors of variability.

Since the studies of dietary intervention and miRNA analysis are reduced to only one study for humans [93] and another for mice [94], where the miRNAs considered are different, it is not possible to establish whether there are similarities. However, there are miRNAs that seem to be detected both at the circulating level in patients and in the brain of AD mice following a PA intervention and, moreover, exhibit similar behavior. This is the case for the upregulation of miR-129-5p [100] and the downregulation of miR-192-5p [99]. Moreover, these miRNAs, that have been reported to affect the same downstream pathways related to the pro-inflammatory response associated with AD, emerge as promising biomarkers in this context.

Other PA intervention studies have found another group of promising miRNAs, which are those that have shown a common response in several studies, even in different murine models of the pathology. Unfortunately, they have only been analyzed in mouse brain, but not in humans. This is the case of miR-15b-5p and miR-148b-3p, overexpressed after exercise [103,105]. In relation to the pathology, miR-15b has been reported to be associated with ageing [132], but there are no data about the role of miR-148b-3p in this disease [105]. Another example to consider is miR-130a-3p, since its expression is upregulated after endurance exercise intervention in two different AD mouse models [101,103]. In fact, miR-130a-3p is expressed at a low level in the AD mice; therefore, its upregulation by exercise could benefit the recovery of the cognitive situation [133]. On the other hand, miR-132 is also downregulated after exercise at brain level in AD mice in two distinct mouse models [103,104]. This miRNA seems to play an effect on the central nervous system [134]. Therefore, these brain tissue miRNAs that respond to PA intervention in AD mouse models are exciting candidates for consideration for analysis in human studies. Nevertheless, translating the results observed in AD murine models to patients can be complex, since miRNA responses may be different between the two species. To overcome this, a useful approach may be to focus on the physiological pathways that are affected in human studies, as we have performed in the pathway enrichment analysis, and to determine in AD murine models whether these same pathways are affected by the disease. This could facilitate finding new targets for therapeutic intervention.

Summarizing, even though lifestyle interventions have been shown to have a positive effect on several metabolic pathways dysregulated during AD, in this review we have been able to ascertain that, to date, only a few miRNAs have been found to be involved in these cellular processes and to respond to lifestyle modifications (Figure 5). As we have emphasized, there are still many limitations to further deepen the understanding of how miRNAs may be participating in the physiological pathways altered by AD. Thus, more studies are needed to overcome these limitations, which, once surpassed, will allow even more exhaustive studies to be carried out. It is also important not to lose sight of the need to consider lifestyle as complex combination of factors that act together, such as diet and exercise, in the modulation of miRNAs, which have the potential to be used both as epigenetic biomarkers and as therapeutic targets before and during the disease.

## 6. Conclusions

The literature review conducted in this article supports the conclusion that there are few scientific publications on how lifestyle interventions, through miRNA expression modifications, contribute to reducing the risk of AD or its progression. In fact, in the case of human studies, none have been found that analyze the effect of nutritional interventions on the c-miRNA profile in patients with AD or MCI and only two studies have analyzed how exercise interventions, only considering endurance exercise, modify this profile of c-miRNAs. Furthermore, in these studies we have found important limitations, which we have outlined, that need to be overcome in future publications.

Moreover, when addressing miRNA modifications associated with a multifactorial pathology such as LOAD, it is necessary to use broader miRNA analytical approaches, as most of the available studies only analyze a very small panel of miRNAs previously associated with this pathology. This limits the discovery capacity of the analysis and, therefore, the identification of new biomarkers and potential therapeutic targets.

Despite these limitations, from the available literature some miRNAs emerge as promising biomarkers, such as let-7g-5p, miR-107, and miR-144-3p in relation to overall diet quality and miR-129-5p, miR-192-5p, miR-15b-5p, miR-148b-3p, miR-130a-3p, and miR-132 in relation to aerobic exercise training.

These results support the need to further explore this field of AD epigenetics, as it is suggested that lifestyle-related miRNA changes may have a very relevant role as biomarkers and as therapeutic targets during the disease.

## Figures and Tables

**Figure 1 nutrients-15-03688-f001:**
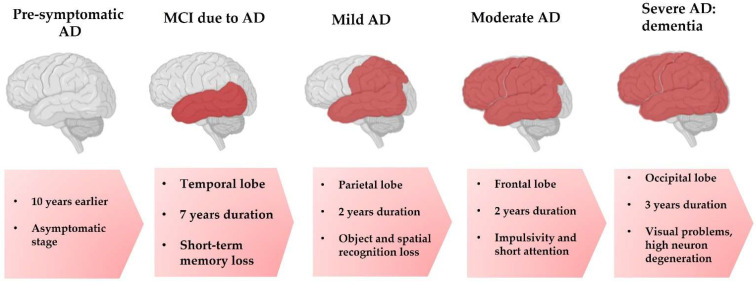
Progressive stages in the development of AD.

**Figure 2 nutrients-15-03688-f002:**
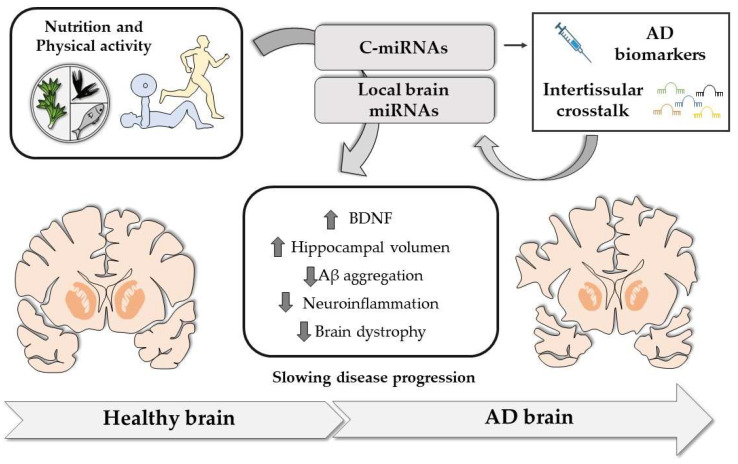
Lifestyle interventions such as food and nutrient intake and physical activity in early pre-symptomatic AD can regulate the expression of both circulating microRNAs (c-miRNAs) and brain-derived miRNAs. In the case of c-miRNAs, they could be used as early biomarkers of disease risk or could create an inter-tissular crosstalk by regulating the expression of distant genes in different tissues, such as the brain. Regarding brain miRNAs, some of them are involved in modulating dysregulated pathways in AD pathophysiology, increasing brain-derived neurotrophic factor (BDNF) levels and hippocampal volume, and reducing Aβ aggregation, neuroinflammation, and brain atrophy, which ultimately could delay the development of the disease.

**Figure 3 nutrients-15-03688-f003:**
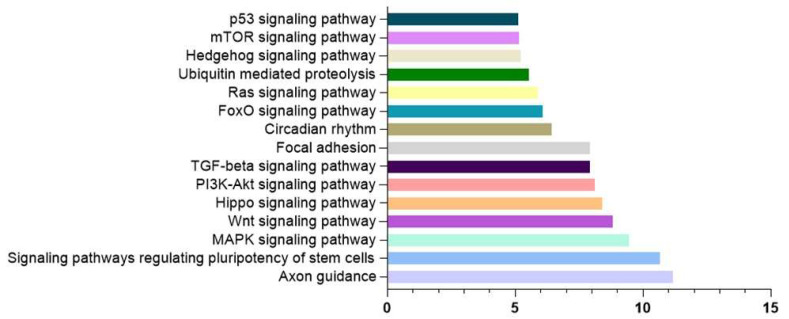
In silico analysis of let-7g-5p, miR-107, and miR-144-3p common targets. KEGG analysis based on TargetScan predicted targets on miRpath v4. Data are presented as -log_10_ of *p*-value.

**Figure 4 nutrients-15-03688-f004:**
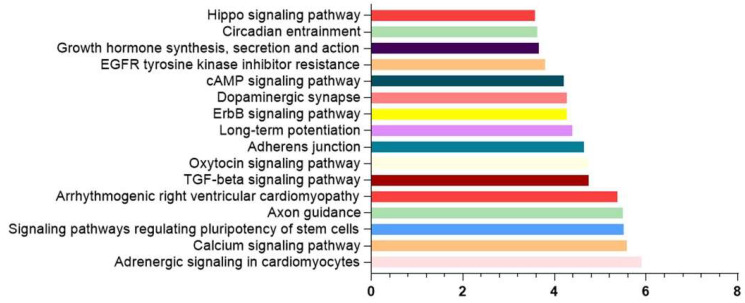
In silico analysis of miR-129-5p targets. KEGG analysis based on TargetScan predicted targets on miRpath v4. Data are presented as -log_10_ of *p*-value.

**Figure 5 nutrients-15-03688-f005:**
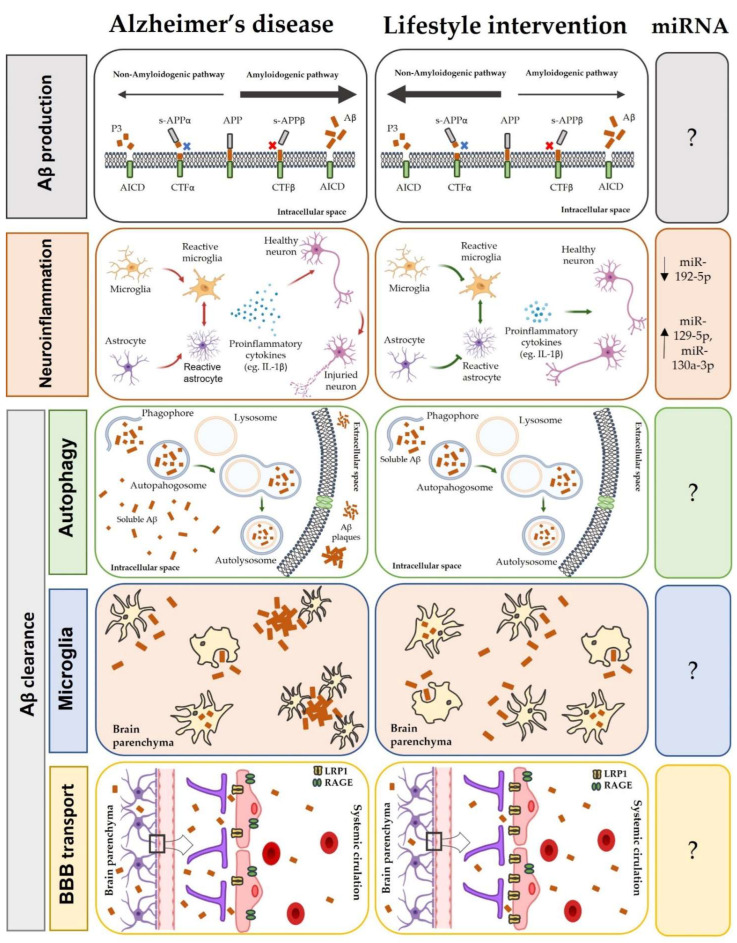
Lifestyle interventions effect on the modulation of miRNAs involved in the pathophysiology of AD. In relation to β-amyloid production, the healthy lifestyle will promote the non-amyloidogenic pathway, favoring the release of P3 peptide, which has neuroprotective effects. In relation to neuroinflammation, activation of microglia and astrocytes will be reduced, releasing fewer pro-inflammatory cytokines that preserve a healthy state for neurons. In this metabolic pathway, exercise can reduce miR-192-5p expression and increase miR-129-5p and miR-130a-3p expression, reducing neuroinflammation. In terms of β-amyloid clearance, lifestyle interventions can enhance autophagic activity and effectively clear the neuronal β-amyloid burden. Furthermore, they can increase phagocytosis of Aβ by microglia into soluble forms and promote outward transport of Aβ into the peripheral system at an increased rate due to increased expression of LRP1 transporters at the blood–brain barrier.

**Table 1 nutrients-15-03688-t001:** Summary of some of the effects of several dietary compounds in Alzheimer’s disease previously described.

Diet Compounds	Positive Effects	Negative Effects
Polyphenolic compounds [20,25]	↑ Aβ aggregation	–
	↑ BDNF	
Omega-3 fatty acids [26,27,28]	↑ LRP1 expression	–
	↑ Autophagy	–
	↓ Neuroinflammation	–
	↓ Neuronal loss	–
Cholesterol-rich diets [29]	–	↑ Aβ aggregation
Red and processed meat [29]		

Legend: Aβ, β–amyloid peptide; ↑, increase; ↓, decrease; –, not described.

**Table 2 nutrients-15-03688-t002:** Summary of murine models of Alzheimer’s disease used to study microRNAs in the brain.

AD Mouse Model *	Genetic Modification	Behavioral Phenotype	Histopathology
APP/PS1	Double transgenic for two human known mutations in: *APP* gene: K670_M671delinsNL (Swedish mutation) *PSEN1* gene: L166P Mouse promoter: mouse prion (central nervous system)	6 months: contextual memory impairment onset. 12 months: spatial memory impairment onset.	6 months: Aβ accumulation onset. 9 months: Aβ plaques in hippocampus and cerebral cortex.
5xFAD	Double transgenic for five human known mutations in: *APP* gene (three mutations): K670_M671delinsNL (Swedish mutation); I716V (Florida); V717I (London) *PSEN1* gene (two mutations): M146L; L286V Mouse promoter: Thy1 (neurons)	3–6 months: spatial memory impairment onset. 9 months: sensorimotor deficits.	2 months: Aβ plaques in hippocampus, cerebral cortex, thalamus, and spinal cord.
3xTgAD	Triple transgenic for three human known mutations in: *APP* gene: K670_M671delinsNL (Swedish mutation) *PSEN1* gene: M146V *MAPT* gene: P301L Mouse promoter: Thy1.2 (neurons)	4 months: retention deficits onset. 6 months: spatial learning and memory impairment.	6 months: Aβ plaques in frontal cortex, progress with age. 12 months: tau reactivity in hippocampus, progressing to cerebral cortex.
SAMP8	Senescence accelerated mouse-prone 8. Spontaneous phenotype of accelerated ageing	2 months: spatial memory impairment onset.	4 months: Aβ accumulation onset. It is increased with age, but no plaques are observed.

* Additional information regarding AD mouse models can be found at www.alzforum.org (accessed on 16 August 2023). Legend: AD, Alzheimer’s disease; APP, amyloid precursor protein; PSEN1, presenilin–1; MAPT, microtubule–associated protein tau.

**Table 3 nutrients-15-03688-t003:** Studies examining the relationship between dietary patterns or nutrient supplementation on circulating microRNA (miRNA) levels or brain miRNA expression regulation in mild cognitive impairment (MCI) patients or Alzheimer’s disease (AD) murine models.

Reference	Subjects or Animals	AD Diagnosis or AD Mouse Model	Dietary Assessment or Intervention	Type of Sample	miRNAs Analyzed and Method	Change in miRNA Expression
	Humans					
Zhang et al., 2021 [93]	75 MCI patients (62 ± 4.1 years, 36 males and 39 females) and 52 healthy controls (62.5 ± 4.0 years, 24 males and 28 females)	MCI due to AD was diagnosed by neurologists based on cognitive screening tests of Mini-Mental State Examination (MMSE) and Montreal Cognitive Assessment (MoCA)	Diet quality was recorded by the Chinese Dietary Guidelines Index 2018 (CDGI-2018), the Energy-Adjusted Dietary Inflammatory Index (E-DII), and the Healthy Lifestyle Score (HLS)	Serum	hsa-let-7g-5p, hsa-miR-107, hsa-miR-144-3p, and hsa-miR-186-3p by qPCR	hsa-miR-144-3p was higher in MCI patients compared to healthy controls hsa-let-7g-5p, hsa-miR-107, and hsa-miR-186-3p was lower in MCI patients compared to healthy controls
	Animal models					
Liu et al., 2015 [94]	7-month-old APP/PS1 male mice were randomly divided into two groups (n = 6 per group): (1) folic-acid-deficient diet (AD + FD) and (2) control diet (normal folic acid content) (AD + FN)	APP/PS1 *	For 60 days, AD + FD mice received a folic-acid-deficient diet (folic acid content: 0.2 mg/kg diet) and AD + FN mice received a control diet (folic acid content: 2.1 mg/kg diet)	Whole brain	769 miRNAs were examined using mouse miRNome qPCR panels I and II (Exiqon) and subsequent validation by qPCR of differentially expressed mmu-miR-200b-3p, mmu-miR-106a-5p, and mmu-miR-339-5p	mmu-miR-106a-5p, mmu-miR-200b-3p and mmu-miR-339-5p were downregulated in AD + FD mice

* See Table 2 for more information.

**Table 4 nutrients-15-03688-t004:** Studies examining the effect of physical activity (PA) interventions on circulating microRNA (c-miRNA) levels or brain miRNA expression in Alzheimer’s disease (AD) patients or AD murine models.

Reference	Subjects or Animals	AD Diagnosis or AD Mouse Model	Training Protocol	Type of Sample	miRNAs Analyzed and Method	Change in miRNA Expression
	Humans					
Qin et al., 2021 [99]	90 AD patients divided in 2 groups: Exercise group (n = 45, 61.89 ± 6.38 years, 17 males and 28 females) and Control group (n = 45, 61.78 ± 7.06 years, 15 males and 30 females)	Neurological Disorders and Stroke-Alzheimer Disease and Related Disorders (NINCDS–ADRDA) diagnosis criteria	Cycling training at 70% of maximum heart rate for 3 months	Serum	hsa-miR-192-5p by qPCR	hsa-miR-192-5p was lower in the serum of AD patients in the exercise group compared to AD sedentary patients
Li et al., 2020 [100]	80 AD patients divided in 2 groups: Exercise group (n = 40, 73.05 ± 7.11 years, 17 males and 23 females) and Control group (n = 40, 72.45 ± 7.28 years, 16 males and 24 females)	NINCDS–ADRDA diagnosis criteria	Cycling training at 70% of maximum heart rate for 3 months	Serum	hsa-miR-129-5p by qPCR	hsa-miR-129-5p was higher in the serum of AD patients in the exercise group compared to AD sedentary patients
	Animal models					
Lu et al., 2022 [101]	8-month-old double transgenic APP/PS1 mice divided in 2 groups (n = 12 per group): Sedentary and Voluntary exercise	APP/PS1 *	Voluntary wheel running for 8 weeks, 7 days/week	Hippocampus	mmu-miR-130a-3p by qPCR	mmu-miR-130a-3p was higher in the APP/PS1 mice in the voluntary exercise group compared to sedentary mice group
Qin et al., 2021 [99]	8-month-old double transgenic APP/PS1 mice divided in 2 groups (n = 12 per group): Control and Voluntary exercise	APP/PS1 *	Voluntary wheel running for 4 weeks	Hippocampus	mmu-miR-192-5p by qPCR	mmu-miR-192-5p was lower in the APP/PS1 mice in the voluntary exercise group compared to sedentary mice group
Li et al., 2020 [100]	8-month-old double transgenic APP/PS1 mice divided in 2 groups (n = 12 per group): Control and Voluntary exercise,	APP/PS1 *	Voluntary wheel running for 4 weeks	Hippocampus	mmu-miR-129-5p by qPCR	mmu-miR-129-5p was higher in the APP/PS1 mice in the voluntary exercise group compared to sedentary mice group
Shvarts-Serebro et al., 2021 [102]	7-month-old 5xFAD and WT female mice divided into 2 groups (n = 5 per group): Control and Enriched environment (EE).	5xFAD *	EE with running wheels for 8 weeks	Hippocampus	mmu-miR-128 by qPCR	-
Dungan et al., 2020 [103]	2-month-old WT female mice and 3xTg-AD mice were used and divided in 2 groups: Sedentary (n = 6 WT and n = 5 3xTg-AD) and Exercise group (n = 5 3xTg)	3xTg-AD *	Progressive weighted wheel running (PoWer) for 20 weeks	Hippocampus	Nanostring nCounter miRNA Expression Panels and subsequent validation by qPCR	mmu-miR-29a, mmu-miR-29b, mmu-miR-29c, mmu-miR-107, mmu-miR-328, mmu-miR-129, mmu-miR-140, mmu-miR-148b, mmu-miR-15b-5p, and mmu-miR-130a were higher and mmu-miR-98 and mmu-miR-132 were lower in the 3xTg-AD mice in the exercise group compared to sedentary mice group.
Dong et al., 2018 [104]	6-month-old WT males (SAMR1) and SAMP8 mice were divided in 2 groups: Sedentary (n = 10 SAMR1 and n = 10 SAMP8) and Exercise (n = 10 SAMR1 and n = 10 SAMP8)	SAMP8 *	Voluntary wheel running for 4 weeks	Cortex, Hypothalamus, Striatum, and Hippocampus	mmu-miR-132 by qPCR	mmu-miR-132 was lower in the SAMP8 mice in the exercise group compared to sedentary mice group
Cosín-Tomás et al., 2014 [105]	8-month-old WT females (SAMR1) and SAMP8 mice were divided in 2 groups: Sedentary (n = 4 SAMR1 and n = 4 SAMP8) and Exercise (n = 4 SAMR1 and n = 4 SAMP8)	SAMP8 *	Voluntary wheel running for 8 weeks, 7 days/week	Hippocampus	84 mature miRNAs (miScript^®^miRNA PCR Array) and subsequent validation by qPCR	mmu-miR-28-5p, mmu-miR-98-5p, mmu-miR-148b-3p, mmu-miR-7a-5p, and mmu-miR-15b-5p were higher and mmu-miR-105 and mmu-miR-133b-3p were lower in the SAMP8 mice in the exercise group compared to sedentary mice group.

Legend: n, number of subjects. * See Table 2 for more information on AD mouse models.

## Data Availability

The data presented in this study are available in Figure 3 and Figure 4.

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
