# Peer review of "Modulation of microRNAs through Lifestyle Changes in Alzheimer’s Disease"

_nutrients, 2023, doi:10.3390/nu15173688_

Round 1

Reviewer 1 Report

This is an excellent review.

I only found a couple of trivial grammatical errors. 

Line 414 Should say "only a few studies"

Line 434 says ":miRNAs at the circulating" What does this mean?

Author Response

We thank the reviewer for the constructive revision of the manuscript. This has enabled us to correct the grammatical errors found in the text, which are cited below: 

Point 1: Line 414 Should say "only a few studies" 

Response 1: It has been corrected in the text, line 441 in the new version of the manuscript.  

Point 2: Line 434 says ":miRNAs at the circulating" What does this mean? 

Response 2: We have corrected the sentence to: “these dysregulated miRNAs at plasma or serum level”, line 461 in the new version of the manuscript.  

Reviewer 2 Report

I would like to extend my congratulations to the authors for their dedicated and insightful work on this research article. There are a few comments on potential improvements:

Abstract:

it lacks clarity on the methodologies used in the reviewed studies

the abstract could benefit from highlighting the potential clinical implications of the identified microRNAs as  targets in AD, enhancing its significance for both researchers and clinicians

*Introduction

The introduction provides a comprehensive overview of the subject.

However, the text could be improved by clearly stating the objectives of the review or study that will follow

*Regulation of altered miRNAs in AD pathophysiology by diet

The studies 93-94 would benefit from more in-depth explanations of experimental methodologies, clearer presentation of results, and contextualization of findings within the broader AD research landscape.

*Exercise modulates the expression of miRNAs altered in AD

Human and Mouse models are miss-sign as 2.1 and 2.2. Please correct it

Regarding the human studies, the descriptions of the two studies analyzing the effect of endurance exercise on serum c-miRNA levels in AD patients are clear. However, more comprehensive details on the exercise protocols, such as frequency, duration, and intensity, are needed. These details are important as they can significantly influence miRNA expression profiles. Additionally, the absence of a matched control group of healthy subjects in these studies is a limitation that should be highlighted.

The section on mouse models provides a comprehensive overview of studies that have investigated exercise-induced miRNA expression changes. To enhance clarity, a clearer distinction between the different mouse models and their specific AD-related characteristics could be made

*Pathway enrichment analysis of miRNAs modulated by lifestyle interventions during AD in humans

While the analysis and discussion of metabolic pathways associated with specific miRNAs (let-7g-5p, miR-107, miR-144-3p, miR-129-5p, miR-192-5p) are well-presented, it is essential to clarify whether these pathways are directly implicated in AD pathology or if they intersect with other related conditions. Providing more explicit links to AD pathology could strengthen the relevance of the findings.

while mentioning the potential impact of dietary interventions on miRNA expression, it would be prudent to acknowledge the complexity of dietary effects and how multiple variables could influence miRNA regulation. This will prevent overgeneralization of results.

*Discussion

The authors correctly acknowledge the limited data in this field, but the discussion could be enriched by emphasizing the importance of these gaps in knowledge. This would highlight the research directions that need to be pursued, particularly the need for more comprehensive studies encompassing multiple lifestyle factors, such as diet and exercise.

It's important to discuss the challenges of translational research from murine models to humans. This could involve addressing the potential discrepancies in miRNA responses between murine models and human patients and considering the unique physiological and genetic aspects of human AD patients.

*Conclusions

The statement that "there are not yet enough scientific publications" could be refined to provide a more specific indication of the extent of the gap in research

*References:

Please review style of references. 129 is missing

Author Response

We thank the reviewer for the valuable comments raised which have helped us to enhance the quality and comprehension of our manuscript. Below you can find a detailed response addressing point-by-point your insightful suggestions:

Abstract:

Point 1: it lacks clarity on the methodologies used in the reviewed studies

Response 1: We thank the reviewer for this helpful comment. Due to space limitations in the abstract, we originally minimised the information on the methodology used in the selected studies. However, as the reviewer points out, this information is very relevant and we have modified the abstract accordingly, adding the gaps and limitations, mainly methodological, that we have found in the literature included in this review.

Point 2: the abstract could benefit from highlighting the potential clinical implications of the identified microRNAs as targets in AD, enhancing its significance for both researchers and clinicians

Response 2: This is a very pertinent comment and we have modified the abstract accordingly.

*Introduction

Point 3: The introduction provides a comprehensive overview of the subject. However, the text could be improved by clearly stating the objectives of the review or study that will follow

Response 3: The objective of the study has been inserted at the end of section 1. Introduction (lines 232-240): “Therefore, the aim of this review has been to compile and to critically analyze the available information on the modulatory effect of diet and PA on the expression of miRNAs described as dysregulated in the context of AD. In this way, we will be able to establish critical points for further progress in the development of strategies to: i) explore the value of miRNAs as prognostic biomarkers in MCI and AD, taking into account the modulatory effect of lifestyle; ii) implement non-pharmacological, cost-effective, targeted and personalized interventions, whose effect on patients' risk trajectories can be monitored; and iii) identify new therapeutic targets that allow the development of new treatments for this devastating disease.”

*Regulation of altered miRNAs in AD pathophysiology by diet

Point 4: The studies 93-94 would benefit from more in-depth explanations of experimental methodologies, clearer presentation of results, and contextualization of findings within the broader AD research landscape.

Response 4: We have tried to go deeper into the studies you refer to in your comment, especially in study 94, by expanding the information in the text (sections 2.1 and 2.2, respectively). The contextualization of these results within the AD research landscape has been included in the Discussion section. There, we have tried to unify the limitations of the studies described in sections 2 and 3, as many of them are common.

*Exercise modulates the expression of miRNAs altered in AD

Point 5: Human and Mouse models are miss-sign as 2.1 and 2.2. Please correct it

Response 5: The reviewer is right. It has been corrected in the text.

Point 6: Regarding the human studies, the descriptions of the two studies analyzing the effect of endurance exercise on serum c-miRNA levels in AD patients are clear. However, more comprehensive details on the exercise protocols, such as frequency, duration, and intensity, are needed. These details are important as they can significantly influence miRNA expression profiles. Additionally, the absence of a matched control group of healthy subjects in these studies is a limitation that should be highlighted.

Response 6: We absolutely agree with the reviewer. The absence of the details of the training protocols and the absence of healthy control subjects in both studies is something that also caught our attention.

Regarding the details of the training protocol, the authors do not provide any additional information beyond what we have included both in text and in Table 2, i.e., a cycling training at 70% of maximum heart rate for 3 months. We have reflected this limitation in the text (lines 354-356): “In both cases, the training protocol was based on three months of cycling at a maximum heart rate of 70% [98,99], with no further data on how this protocol was implemented.”

As for the other limitation, the absence of a matched control group of healthy subjects, we have highlighted this statement in the text between lines 360 and 362: “Additional weaknesses in the experimental design were found. First, the lack of a matched control group of healthy subjects. Both studies included only AD patients that were divided into two groups: exercise and non-exercise (Table 4).”

Point 7: The section on mouse models provides a comprehensive overview of studies that have investigated exercise-induced miRNA expression changes. To enhance clarity, a clearer distinction between the different mouse models and their specific AD-related characteristics could be made.

Response 7: We have included a description of the AD murine models used for miRNA studies as Table 2 in the new version of the manuscript. It was added in the Introduction section when the use of mouse models is first mentioned. The data were collected from the repository Alzforum.

*Pathway enrichment analysis of miRNAs modulated by lifestyle interventions during AD in humans

Point 8: While the analysis and discussion of metabolic pathways associated with specific miRNAs (let-7g-5p, miR-107, miR-144-3p, miR-129-5p, miR-192-5p) are well-presented, it is essential to clarify whether these pathways are directly implicated in AD pathology or if they intersect with other related conditions. Providing more explicit links to AD pathology could strengthen the relevance of the findings.

Response 8: We have added in the text complementary information on the role of these pathways in AD pathology, trying to reinforce the relevance of the findings. This new text can be found in lines 492-510.

Point 9: while mentioning the potential impact of dietary interventions on miRNA expression, it would be prudent to acknowledge the complexity of dietary effects and how multiple variables could influence miRNA regulation. This will prevent overgeneralization of results.

Response 9: We have tried to reflect this excellent suggestion in a paragraph that appears on the lines 573-577: “However, due to the complexity of dietary effects and the fact that miRNAs can be regulated by multiple factors, it is difficult to isolate the interaction between them. For this reason, it is necessary to approach the understanding of the effect of diet on miRNA expression from different perspectives, including not only studies in humans or murine models, but also cellular models.”

*Discussion

Point 10: The authors correctly acknowledge the limited data in this field, but the discussion could be enriched by emphasizing the importance of these gaps in knowledge. This would highlight the research directions that need to be pursued, particularly the need for more comprehensive studies encompassing multiple lifestyle factors, such as diet and exercise.

Response 10: We have tried to enrich the discussion with your suggestions in the text (lines 674-684): “Summarizing, even though lifestyle interventions have been shown to have a positive effect on several metabolic pathways dysregulated during AD, in this review we have been able to ascertain that, to date, only a few miRNAs have been found to be involved in these cellular processes and to respond to lifestyle modifications (Figure 5). As we have emphasized, there are still many limitations to further deepen the understanding of how miRNAs may be participating in the physiological pathways altered by AD. Thus, more studies are needed to overcome these limitations, which, once surpassed, will allow even more exhaustive studies to be carried out. It is also important not to lose sight of the need to consider lifestyle as complex combination of factors that act together, such as diet and exercise, in the modulation of miRNAs, which have the potential to be used both as epigenetic biomarkers and as therapeutic targets before and during the disease.”

Point 11: It's important to discuss the challenges of translational research from murine models to humans. This could involve addressing the potential discrepancies in miRNA responses between murine models and human patients and considering the unique physiological and genetic aspects of human AD patients.

Response 11: We have included this valuable suggestion in a paragraph that appears on lines 672-678: “Nevertheless, translating the results observed in AD murine models to patients can be complex, since miRNAs responses may be different between the two species. To overcome this, an useful approach may be to focus on the physiological pathways that are affected in human studies, as we have performed in the pathway enrichment analysis, and to de-termine in AD murine models whether these same pathways are affected by the disease. This could facilitate finding new targets for therapeutic intervention.”

*Conclusions

Point 12: The statement that "there are not yet enough scientific publications" could be refined to provide a more specific indication of the extent of the gap in research

Response 12: We have tried to refine the information on lines 692-704 in the new version of the manuscript: “The literature review conducted in this article supports the conclusion that there is little scientific publications on how lifestyle interventions, through miRNA expression modifications, contribute to reducing the risk of AD or its progression. In fact, in the case of human studies, none have been found that analyze the effect of nutritional interventions on the c-miRNA profile in patients with AD or MCI and only two studies have analyzed how exercise interventions, only considering endurance exercise, modify this profile of c-miRNAs. Furthermore, in these studies we have found important limitations, which we have outlined, that need to be overcome in future publications.

Moreover, when addressing miRNA modifications associated with a multifactorial pathology such as LOAD, it is necessary to use broader miRNA analytical approaches, as most of the available studies only analyze a very small panel of miRNAs previously associated with this pathology. This limits the discovery capacity of the analysis and, therefore, the identification of new biomarkers and potential therapeutic targets.”

*References:

Point 13: Please review style of references. 129 is missing

Response 13: It has been corrected in the text.

Reviewer 3 Report

None

Author Response

We thank the reviewer for his/herthorough review and the helpful suggestions that we have used to improve the quality and comprehension of our manuscript. 

Major question 

Point 1: It would be nice to summarize the association of diet habits with AD pathophysiology in a table.  

Response 1: We particularly welcome this suggestion. Thus, we have included in the Introduction a table summarising the effects of some of the components of the diet on the development of this pathology (Table 1 in the new version of the manuscript). 

Minor questions 

Point 2: Spelling mistakes in Figure2., agregation, dystrophy.  

Response 2: They have been corrected in Figure 2. 

Point 3: Many references are not documented well. Please make modifications. 

Response 3: We appreciate the thorough review of the references. We have made the appropriate corrections to those references which were not properly inserted.   
